materials science/environmental science

water treatment, adsorption, nanocomposite, virus retention

**Author for correspondence:**
Thomas Graule
e-mail: thomas.graule@empa.ch

This article has been edited by the Royal Society of Chemistry, including the commissioning, peer review process and editorial aspects up to the point of acceptance.

# Enhanced virus filtration in hybrid membranes with MWCNT nanocomposite

Zoltán Németh[1,2,3], Gergő Péter Szekeres[1,2], Mateusz Schabikowski[1,4], Krisztina Schrantz[1,5], Jacqueline Traber[6], Wouter Pronk[6], Klára Hernádi[2] and Thomas Graule[1]

[1]Laboratory for High Performance Ceramics, Empa, Swiss Federal Laboratories for Materials Science and Technology, Überlandstrasse 129, Dübendorf 8600, Switzerland
[2]Department of Applied and Environmental Chemistry, University of Szeged, Rerrich Béla tér 1, Szeged 6720, Hungary
[3]Institute of Chemistry, University of Miskolc, Miskolc-Egyetemváros, Miskolc 3515, Hungary
[4]Institute of Nuclear Physics, Polish Academy of Sciences, 31342 Krakow, Poland
[5]Department of Inorganic and Analytical Chemistry, University of Szeged, Dóm tér 7, Szeged 6720, Hungary
[6]Department of Process Engineering, Eawag, Swiss Federal Institute of Aquatic Science and Technology, Überlandstrasse 133, Dübendorf 8600, Switzerland

KH, 0000-0001-9419-689X

Membrane separation is proved to be a powerful tool for several applications such as wastewater treatment or the elimination of various microorganisms from drinking water. In this study, the efficiency of inorganic composite-based multi-walled carbon nanotube (MWCNT) hybrid membranes was investigated in the removal of MS2 bacteriophages from contaminated water. With this object, multi-walled carbon nanotubes were coated with copper(I) oxide, titanium(IV) oxide and iron(III) oxide nanoparticles, respectively, and their virus removal capability was tested in both batch and flow experiments. Considering the possible pH range of drinking water, the filtration tests were carried out at pH 5.0, 7.5 and 9.0 as well. The extent of MS2 removal strongly depended on the pH values for each composite, which can be due to electrostatic interactions between the membrane and the virus. The most efficient removal (greater than or equal to 99.99%) was obtained with the $Cu_2O$-coated MWCNT membrane in the whole pH range. The fabricated nanocomposites were characterized by X-ray diffraction, specific surface area measurement, dynamic light scattering, zeta potential measurement, Raman spectroscopy, transmission electron microscopy and scanning electron microscopy. This study presents a simple route to design novel and effective nanocomposite-based hybrid membranes for virus removal.

# 1. Introduction

One of the main challenges of humanity is the demand for safe drinking water. Based on current records, two billion people are left without sufficient sanitation, and about half of these people lack access to safe drinking water [1]. Over the course of history, waterborne diseases have put humanity several times in danger, and even today, the viral and bacterial contamination of drinking water can cause severe plague outbreaks [2]. Therefore, a technological revolution has started in the field of drinking water sanitation, including the involvement of nanotechnology [3]. The biggest achievements could be made by combining nanotechnology with membrane processes, such as in ultrafiltration membranes [4], composite membranes [5] and photocatalytic membranes [6], but modified ceramic filters [7], activated carbon or carbon nanotubes (CNTs) [8,9] can contribute to a more efficient water purification process as well [10,11].

The outstanding physical properties of CNTs, e.g. their mechanical and chemical durability, as well as their thermal and electrical conductivity [12] allow for the use of CNTs in many applications [13,14]. Furthermore, their affinity to adsorb organic compounds [15] and the high specific surface area [16] indicate their potential use in water purification [17,18], which is further exemplified by their reported antimicrobial activity [19]. In recent studies, CNT-based membranes were shown to be effective in water purification [20], presenting multi-logarithmic extents of antimicrobial retention [21–23]. Even though, in the past years, the use of CNTs has been brought together with environmental and health risks [24–26], comparative studies proved that multi-walled CNTs (MWCNTs) were much less toxic than single-walled CNTs (SWCNTs), because of the differences in diameter and surface chemistry [27–29]. Therefore, in this study, we only used MWCNTs.

As inorganic oxides possess biocidal properties [30–35], the combination of MWCNTs with inorganic oxides can further enhance those properties of MWCNTs. As Montgomery & Elimelech showed in their study, CNTs coated with $TiO_2$, $Fe_2O_3$ and $Cu_2O$ are promising in the adsorption-based water purification [1]. Given the small average size of virions, in the range of tens to a couple of hundreds of nanometres [36], their removal from drinking water is a challenge, and a virus filter with suitable water permeability can therefore mainly be based on adsorption processes [37–39]. In our recent study, a copper-coated nanofibrillated cellulose-based virus filter was discussed, which showed up to 5-log virus removal (adsorption combined with inactivation on Cu surfaces) with MS2 bacteriophages [40]. To the best of our knowledge, only a few publications discuss the antimicrobial activity of MWCNT composite membranes [17,41,42], and virus filtration is in the focus of only a fraction of those, e.g. in the work of Kim *et al.* who demonstrated the virus- and bacterium-removal capacity of MWCNT-Ag nanocomposite membranes in water at low pressure [41].

The main objectives of this study were to design and characterize polytetrafluoroethylene (PTFE) and MWCNT-based composite hybrid membranes, to quantify the efficiency of the hybrid membranes by removing MS2 bacteriophages from aqueous solutions, and to compare the performance of the different MWCNT-based filters. Since the pH of naturally accessible water is usually in the range between 6.5 and 9.5 [2,3], the pH values for filtration tests were set to be 5.0, 7.5 and 9.0, respectively. Nanocomposite-based hybrid membranes were prepared via the coating of MWCNTs with titanium dioxide ($TiO_2$), iron(III) oxide ($\alpha$-$Fe_2O_3$) and copper(I) oxide ($Cu_2O$) adsorbents, respectively, and their deposition onto PTFE membranes. The unique system presented in this study could potentially be used as a disinfection membrane system for antiviral water treatment.

# 2. Material and methods

## 2.1. Materials

Commercial MWCNT was purchased from Nanothinx S.A. (Patra, Greece—NTX1 MWCNT, purity greater than 97%). The physical properties of MWCNTs—according to the technical datasheet—are presented below. The average diameter of the MWCNTs was between 15 and 35 nm, while their length was in the range of 10–30 μm. The specific surface area of the MWCNTs was $110\ m^2\ g^{-1}$. Initially, MWCNT powder was cleaned with hydrochloric acid (HCl, approx. 10 wt%, diluted from approx. 37 wt%, Sigma-Aldrich, Switzerland) to remove the remaining catalyst, including the catalyst particles. After that, the MWCNT was filtered and washed with deionized water, until neutral pH was achieved.

For the composite fabrication, the following precursor compounds were used: titanium isopropoxide ($Ti[OCH(CH_3)_2]_4$), iron(II) chloride tetrahydrate ($FeCl_2 \times 4H_2O$) and copper(II) acetate monohydrate

(Cu[CH$_3$COO]$_2$ × H$_2$O). All the used precursors were purchased from Sigma-Aldrich (Switzerland). The applied solvents were absolute ethanol (EtOH—HPLC grade from Sigma-Aldrich (Switzerland)) and nanopure water, purified by Barnstead™ NANOPure® Diamond device (Thermo Scientific, USA). The as-purified water is simply referred to as 'water' in further discussions. PTFE filters (pore size: 5 µm, diameter: 25 mm, Omnipore—JMWP02500) were used as support to prepare MWCNT-based composite hybrid membranes.

Bacteriological agar, d-glucose, sodium hydroxide (NaOH) and sodium dihydrogen phosphate dihydrate (NaH$_2$PO$_4$ × 2H$_2$O) were purchased from Sigma-Aldrich (Switzerland). Calcium chloride dihydrate (CaCl$_2$ × 2H$_2$O), microbiology yeast extract and glycerol were provided by Merck Eurolab (Switzerland). Streptomycin was purchased from AppliChem PanReac (Germany). Tryptone (Difco 0123) and sodium chloride (NaCl) were purchased from Becton Dickinson and VWR International (Switzerland), respectively. *Escherichia coli* (Migula 1895) Castellani and Chalmers 1919 (DSM no.: 5695) colonies were used as host cells for MS2 bacteriophage multiplication (DSM no.: 13767). Dry *E. coli* pellets and the MS2 phage suspension were purchased from DSMZ (Braunschweig, Germany).

## 2.2. MWCNT-based composite and membrane preparation

The TiO$_2$/MWCNT and α-Fe$_2$O$_3$/MWCNT nanocomposites were synthesized by a facile impregnation method, following our former recipe [43]. Purified MWCNT was the modifying component and for TiO$_2$ and α-Fe$_2$O$_3$ component preparation Ti[OCH(CH$_3$)$_2$]$_4$ and FeCl$_2$ × 4H$_2$O, were used as precursors, respectively. MWCNT content was 10 or 20 wt% of the final crystallized product after annealing. The synthesis procedure for the reference materials (TiO$_2$ and α-Fe$_2$O$_3$) for comparative studies was exactly the same as for the composites, but in the absence of MWCNT.

Cu$_2$O/MWCNT composite samples were fabricated by a modified impregnation method [44]. In this case, the MWCNT content was fixed at 10 or 25 wt%. Cu(CH$_3$COO)$_2$ × H$_2$O precursor was dissolved in deionized water, then 2.5 ml of aqueous ammonia solution (25 wt%) was added dropwise (approx. 0.5 ml min$^{-1}$) under continuous stirring. Calculated amount of MWCNT was suspended in the above-mentioned precursor solution for 24 h. Subsequently, the solid material was separated from the solution and dried under vacuum at 70°C. Finally, as-prepared composites were calcined at 300°C in N$_2$ atmosphere for 2 h in a tube furnace.

For the virus removal experiments, different MWCNT composites were deposited onto a 5 µm pore-sized PTFE membrane by sonication and filtration following the membrane preparation procedure published by Brady *et al.* [22]. During this process, 50 mg MWCNT composites were suspended in 250 ml absolute ethanol then the suspension was sonicated for 5 min and finally allowed to cool down. Exploiting the capability of MWCNT to form 'paper' with ease, we could prepare good-quality membranes from these fibrous materials. The deposition of 15 ml of the MWCNT composite suspension (0.2 mg ml$^{-1}$) was accomplished by vacuum filtration through the PTFE membrane, to achieve a loading of 0.15 mg cm$^{-2}$ on the surface. As a final step, PTFE-based MWCNT composite membranes were air-dried at room temperature for 30 min.

## 2.3. Media preparation and virus propagation

The required media for bacteria and MS2 growth and filtration (such as CaCl$_2$, antibiotic solution, broth, virus dilution buffer (VDB), hard and soft agar) were produced as offered by Pecson *et al.* [45]. MS2 was replicated using *E. coli* and subsequently purified and concentrated in different steps, according to the protocol provided by DSMZ. The enumeration of bacteriophage MS2 was performed by counting the transparent spots on a double-layered agar plate with a white, continuous *E. coli* layer, where *E. coli* acts as a bacterial host. Each transparent spot derived from one active MS2 bacteriophage, which, in further discussion, is referred to as one plaque-forming unit (1 PFU). When it was necessary, logarithmic dilutions of MS2 were prepared to decrease the number of plaques to the interval of 10–100 per plate, thus making the counting easier and less susceptible to mistakes. Because of the sensitive nature of MS2 bacteriophages, the amount of PFUs in a suspension had to be periodically determined. The initial stock of purified MS2 had the concentration of 5 × 10$^6$ PFU ml$^{-1}$. For further usage, including each measurement, the initial stock was diluted in VDB. Hence, the detection limit of approximately 4 LRV (log reduction value) or 99.99% was determined by the phage enumeration that used a maximum sample volume of 2 ml. VDB was prepared using NaH$_2$PO$_4$ × 2H$_2$O, NaCl and water. The pH (pH = 7.5) was adjusted by adding drops of 0.1 M NaOH solution. Room temperature (23°C) was maintained throughout the virus filtration experiments.

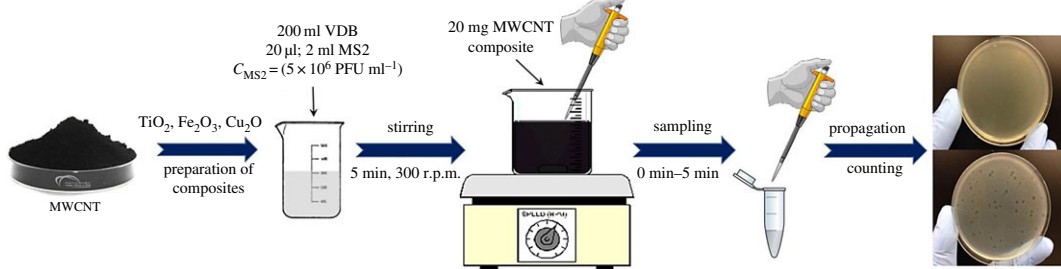

**Figure 1.** Experimental set-up of batch experiments.

## 2.4. Virus adsorption experiments

### 2.4.1. Batch experiments applying composite suspensions

Batch experiments (figure 1) were performed using sterilized 250 ml glass bottles and continuous stirring at 300 r.p.m. by a magnetic stirrer similar to the procedure of our copper-coated cellulose-based hybrid filter [40]. VDB (200 ml; pH: 5.0; 7.5; 9.0) and 20 µl or 2 ml of the $5 \times 10^6$ PFU ml$^{-1}$ virus stock were added into the beaker to investigate 2-Log or 4-Log MS2 adsorption. In order to quantify the filter retention performance, we use the log reduction value (LRV), equation (2.1). The LRV gives a logarithmic expression of the fractional retention ($R$), equation (2.2) [7].

$$LRV_i = -\log10(1 - R_i),\tag{2.1}$$

$$R_i = 1 - \frac{C_i}{C_i(0)}.\tag{2.2}$$

The mixture was then covered with Parafilm® M and stirred for 5 min. Next, 20 mg of MWCNT composite powder was added to the beaker. Samples were taken at t = 0 min (the moment of the addition of MWCNT), and after 1, 2, 3, 4 and 5 min. Then, 100 µl of the purified virus suspension was pipetted into 6–7 ml soft agar at 56°C, with 200 µl bacterium suspension (with the optical density at 640 nm [$OD_{640}$] of 0.2). The mixture was poured onto a hard agar plate and left for solidification. Then, the Petri dish was placed in an incubator at 37°C for 24 h. The concentration of the MS2 bacteriophages was determined after the incubation. Each enumeration of MS2 samples was performed twice, and each condition was tested three times. Every membrane and raw material preparation, as well as the experiments, was performed in the close vicinity of an open flame (propane-butane laboratory torch) to avoid external contamination.

### 2.4.2. Flow experiments applying composite filters

Flow filtration experiments (figure 2) were also performed at three different pH values (pH = 5.0, 7.5 and 9.0, respectively) to cover the whole range of natural water pH variations. Initially, 10 µl, 100 µl or 1 ml of the original virus stock (for 2-Log, 3-Log or 4-Log MS2 adsorption investigations, respectively) was suspended in 100 ml of VDB, covered with Parafilm® M and stirred for 5 min. The mixture was then placed in a sterilized 140 ml plastic syringe with Luer-lock tip (Harvard Apparatus GmbH). In the next step, the MWCNT composite membranes, with a load of 0.15 mg cm$^{-2}$ (3 mg membrane$^{-1}$), were placed into a sterilized swin-lock plastic membrane holder with the diameter of 25 mm (Sigma-Aldrich, Whatman®) and connected to the Luer-lock syringe. The virus retention investigations were performed with the use of a syringe pump (Harvard Apparatus GmbH—PHD UltraTM CP). Two different flow rates were applied during the flow experiments: 5 ml min$^{-1}$ and 10 ml min$^{-1}$. The water flux values were determined using the equations below (equations (2.3)–(2.5)).

$$\text{filter diameter: } 25 \text{ mm} = 0.025 \text{ m} \rightarrow A = r^2\pi = (0.025)^2 \times 3.14 = 1.9625 \times 10^{-3} \text{ m}^2\tag{2.3}$$

$$\text{flow rate: } 5 \text{ ml min}^{-1} = 0.3 \text{ dm}^3 \text{h}^{-1}\tag{2.4}$$

$$\text{flux } (F) = \frac{0.3 \text{ dm}^3}{0.0019625 \text{ m}^2 \text{h}} \approx 150 \text{ dm}^3 \text{m}^{-2} \text{h}^{-1}\tag{2.5}$$

We performed all experiments at a water flux of 150 dm$^3$ m$^{-2}$ h$^{-1}$ (flow rate 5 ml min$^{-1}$, filtration time 20 min) and 300 dm$^3$ m$^{-2}$ h$^{-1}$ (flow rate 10 ml min$^{-1}$, filtration time 10 min) to test the effect of filter approach velocity. Filter permeate samples were collected during filtration into autoclaved tubes

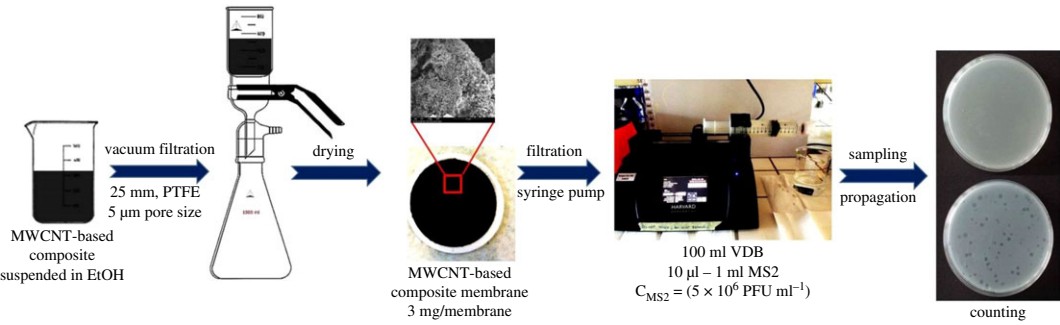

**Figure 2.** Experimental set-up of flow experiments.

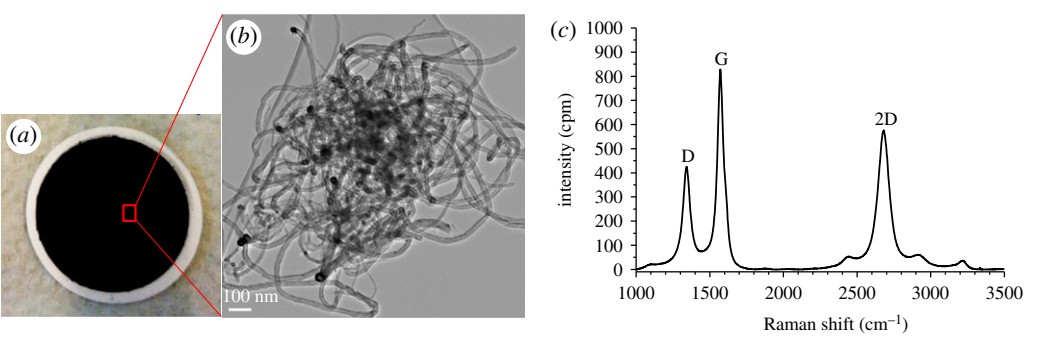

**Figure 3.** The pristine NTX1 MWCNT membrane material: a representative photograph (*a*), a TEM micrograph (*b*) and a Raman spectrum (*c*).

and the virus concentrations were determined by counting the dots in a homogeneous *E. coli* layer, each corresponding to 1 PFU ml$^{-1}$.

## 2.5. Characterization of membranes

For composite membrane characterization, transmission electron microscopy (TEM), scanning electron microscopy (SEM), X-ray powder diffraction (XRD), Raman spectroscopy, specific surface area measurements (BET), Dynamic light scattering (DLS) and zeta potential measurements were performed.

The formation of oxide nanoparticles on the surface of MWCNT was verified by JEOL JEM 2200FS HR-TEM. For TEM investigations, a small amount of the sample was sonicated in 1 ml of distilled water. Then, a few drops of this suspension were dribbled onto the surface of the grid (LC200-Cu TEM grid covered with lacey carbon film, Electron Microscopy Sciences, USA). SEM studies were carried out in an FEI Nova NanoSEM 230 that operated in the 5–15 kV range, after the samples were attached to a conductive carbon tape. The crystalline structure of the as-prepared membrane was determined by powder X-ray diffraction (PANalytical X'Pert Pro MPD machine with a Cu K$\alpha$ ($\lambda$ = 1.5405 Å) radiation). Scanning was performed over a 2$\theta$ range of 10–80° with a step size of 0.0167°. Raman spectroscopy measurements were performed with a Thermo Scientific DXR Raman microscope with a 532 nm laser (5 mW). The specific surface areas of the samples were determined by the adsorption of nitrogen at 77 K according to the method of Brunauer–Emmett–Teller [46]. After the samples were pre-treated at 300°C for 15 min under He atmosphere (50 cm$^3$ min$^{-1}$), measurements were carried out by a single point BET instrument (Beckman-Coulter SA3100). Zeta potential ($\zeta$) measurements were performed by microelectrophoresis (Zetasizer Nano ZS, Malvern Instruments, UK) and streaming potential (Anton-Paar SurPASS) techniques. Clear disposable capillary cells (DTS 1070, Malvern Instruments, UK) were used for the electrophoretic measurements. NaOH and HCl solutions of 0.1 and 0.01 M were used as titrants to adjust the pH values.

# 3. Results and discussion

## 3.1. MWCNT-based nanocomposite characteristics

As an essential characterization of pristine MWCNT, its representative TEM image and Raman spectrum are presented in figure 3. As described in our recent study [47], well-defined bands can be observed at

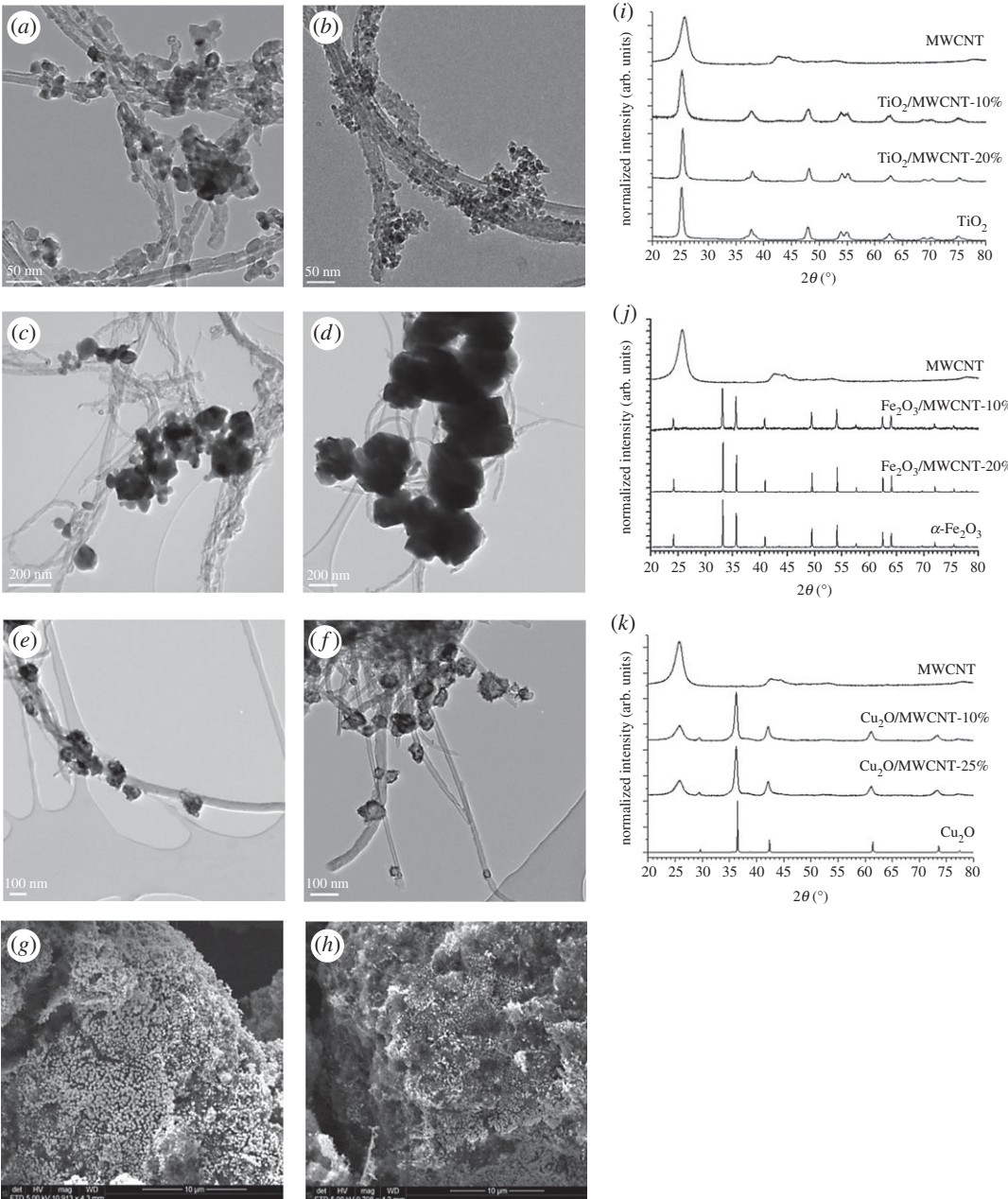

**Figure 4.** Representative TEM (*a–f*) and SEM (*g–h*) micrographs of TiO$_2$/MWCNT (*a,b*), $\alpha$-Fe$_2$O$_3$/MWCNT (*c,d*) and Cu$_2$O/MWCNT (*e–h*) nanocomposite membrane materials with 10 wt% (*a,c,e,g*), 20 wt% (*b,d*) and 25 wt% (*f,h*) MWCNT content, respectively. X-ray diffractograms of raw MWCNT, the treated (400°C—3 h) reference materials (TiO$_2$, Fe$_2$O$_3$, Cu$_2$O) and TiO$_2$/MWCNT (*i*), $\alpha$-Fe$_2$O$_3$/MWCNT (*j*) and Cu$_2$O/MWCNT (*k*) nanocomposite membranes.

1342.7, 1572.2 and 2680.1 cm$^{-1}$ (overtone of D mode), attributing to the D-, G- and 2D-bands of MWCNT, while further weak second-order bands at 2443.9 cm$^{-1}$ (non-dispersive overtone of G), 2917.3 cm$^{-1}$ (longitudinal optic overtone) and 3220.0 cm$^{-1}$ (overtone of G) are also present. Dresselhaus *et al.* [48] also discussed that the appearance of the band at 2443.9 cm$^{-1}$, with its very weak intensity compared to that of the 2700 cm$^{-1}$ band, proposes the high quality of the sample. The D/G band intensity ratio is generally used as an effective indicator for the degree of MWCNT graphitization, where the higher value suggests the presence of more defect sites in the graphitic lattice. The intensity ratios ($I_D$, $I_G$ and $I_{2D}$) between the three main peaks ($I_D/I_G = 0.51$, $I_{2D}/I_G = 0.70$ and $I_D/I_{2D} = 0.74$) testify sp$^2$ structure in our MWCNT sample and approve the high quality and highly graphitic nature of carbon nanotube [49].

The morphology of MWCNT-based composite materials was investigated by TEM and SEM techniques. Figure 4*a–h* shows TEM and SEM micrographs of TiO$_2$/MWCNT (figure 4*a,b*), $\alpha$-Fe$_2$O$_3$/MWCNT

**Table 1.** Particle size and specific surface area of raw and MWCNT-based nancomposite materials.

| sample | $d_{av(TEM)}$ (nm) | $d_{av(XRD)}$ (nm) | BET ($m^2 g^{-1}$) |
|---|---|---|---|
| MWCNT | $25 \pm 10$ | 38 | 110.1 |
| $TiO_2$ (anatase) | $20 \pm 6$ | 25 | 62.7 |
| $\alpha$-$Fe_2O_3$ | $55 \pm 27$ | 64 | 5.3 |
| $Cu_2O$ | $25 \pm 7$ | 30 | 58.9 |
| $TiO_2$/MWCNT-10% | $20 \pm 6$ (TiO$_2$) | 24 | 68.4 |
| $TiO_2$/MWCNT-20% | $25 \pm 8$ (TiO$_2$) | 29 | 82.8 |
| $Fe_2O_3$/MWCNT-10% | $89 \pm 34$ (Fe$_2$O$_3$) | 96 | 6.7 |
| $Fe_2O_3$/MWCNT-20% | $143 \pm 48$ (Fe$_2$O$_3$) | 169 | 20.8 |
| $Cu_2O$/MWCNT-10% | $20 \pm 6$ (Cu$_2$O) | 22 | 85.7 |
| $Cu_2O$/MWCNT-25% | $26 \pm 7$ (Cu$_2$O) | 28 | 96.9 |

(figure 4c,d) and $Cu_2O$/MWCNT (figure 4e–h) nanocomposites at various magnifications, respectively. From EM images, it can be concluded that inorganic nanoparticles ($TiO_2$, $\alpha$-$Fe_2O_3$ and $Cu_2O$) are attached to the surface of MWCNTs. Based on detailed electron microscopy investigation, no significant difference was found between the morphologies of nanocomposites produced with various MWCNT contents.

The crystallinity of heat-treated composite samples was verified by the X-ray diffractometry method. XRD patterns of both pristine components and nanocomposite samples are summarized in figure 4i–k. While diffraction peak at $2\theta = 26.5°$ belongs to the 002 reflection of MWCNT, other diffraction peaks in the range of $20° < 2\theta < 80°$ correspond to the (101), (004), (200), (105), (211), (204), (116) and (220) reflections of anatase in $TiO_2$-containing samples [50]. In the case of $\alpha$-$Fe_2O_3$/MWCNT nanocomposite powder (figure 4j), the Miller indices of $\alpha$-$Fe_2O_3$ are (012), (104), (110), (113), (024), (116), (122), (214) and (300), respectively, as described earlier [51]. Diffractograms in figure 4k illustrate the XRD analysis of $Cu_2O$/MWCNT nanocomposites. It was found that their characteristic reflections were in good correlation with that of pure $Cu_2O$, suggesting that neither impurities nor different oxidation stages of Cu appear in the MWCNT-containing products. Diffraction peaks in the range of $20° < 2\theta < 80°$ correspond solely to the (111), (200) and (220) reflections of $Cu_2O$ [52]. The results of XRD analysis combined with the electron microscopy studies confirmed that the preparation of the $Cu_2O$/MWCNT nanocomposite was done successfully without the damage of $Cu_2O$ phase which was a crucial issue during fabrication.

Using the Scherrer equation [53], the average crystallite size of primary inorganic particles was also determined by X-ray diffractograms (figure 4i–k) (see equation (3.1)). Explaining this well-known equation, D is the diameter in nanometre of the grain or the layer, K is the shape factor (0.89), $\lambda$ is the X-ray wavelength of Cu K$\alpha$ (0.154 nm in the instrument used), $\beta$ is the experimental full-width half maximum of the respective diffraction peak(s) and $\Theta$ is the Bragg angle.

$$D = \frac{K\lambda}{\beta \cos \Theta},$$
(3.1)

Furthermore, the average particle sizes were calculated from the analysis of the TEM images, too, using iTEM software (Olympus Soft Imaging Solutions). The particle size distribution was determined by measuring the size of 100 particles in the case of all samples. We also took into consideration that the TEM images show only a two-dimensional projection of the real three-dimensional particles; consequently, the observed particle size distribution is practically a distribution of the projected dimension of the particles. Average particle size values attained with the two different calculation methods showed good agreement (see also figure 4). In table 1, the average particle diameters ($d_{av}$) calculated from TEM and XRD investigations are summarized for each material. The as-prepared MWCNT-based filter materials were also characterized by $N_2$ adsorption technique to measure their specific surface area (table 1). From data in table 1, it can be concluded that the presence of MWCNT did not significantly affect the average particle sizes of inorganic components during the nanocomposite fabrication procedure, except for $\alpha$-$Fe_2O_3$, which suffered a considerable aggregation and resulted in two to three times bigger particles in the composite. This phenomenon might be

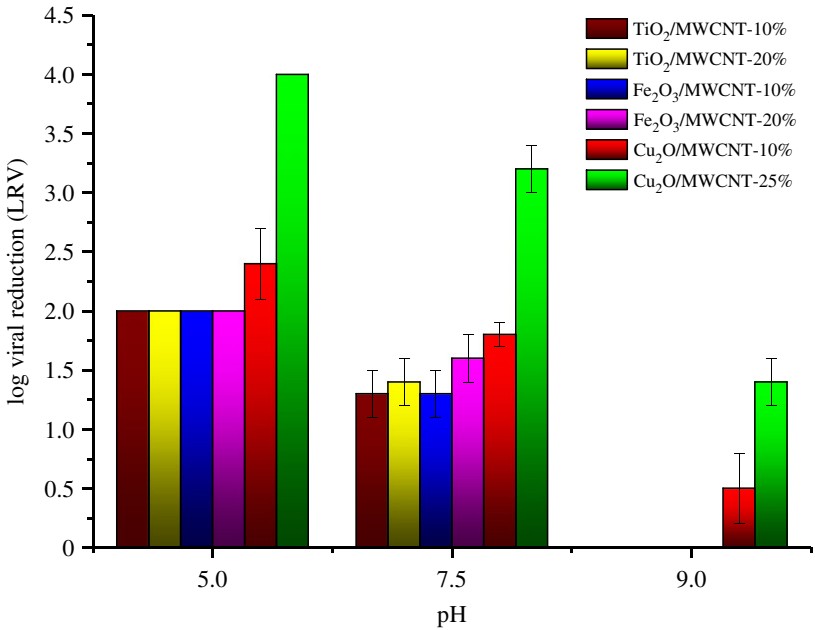

**Figure 5.** MS2 retention of the MWCNT-based nanocomposite membranes in batch experiments.

**Table 2.** MS2 bacteriophage removal efficiency of MWCNT-based nanocomposites at varying pH values in batch experiments.

| sample | LRV—pH 5.0 | LRV—pH 7.5 | LRV—pH 9.0 |
|---|---|---|---|
| MWCNT | 2.0 ± 0.0 log | 1.1 ± 0.3 log | 0.0 ± 0.0 log |
| $TiO_2$ | 2.0 ± 0.0 log | 1.5 ± 0.2 log | 0.0 ± 0.0 log |
| $Fe_2O_3$ | 2.0 ± 0.0 log | 1.7 ± 0.2 log | 0.5 ± 0.2 log |
| $Cu_2O$ | 4.0 ± 0.0 log | 3.9 ± 0.1 log | 3.7 ± 0.2 log |
| $TiO_2$/MWCNT-10% | 2.0 ± 0.0 log | 1.3 ± 0.3 log | 0.0 ± 0.0 log |
| $TiO_2$/MWCNT-20% | 2.0 ± 0.0 log | 1.4 ± 0.2 log | 0.0 ± 0.0 log |
| $Fe_2O_3$/MWCNT-10% | 2.0 ± 0.0 log | 1.3 ± 0.2 log | 0.0 ± 0.0 log |
| $Fe_2O_3$/MWCNT-20% | 2.0 ± 0.0 log | 1.6 ± 0.2 log | 0.0 ± 0.0 log |
| $Cu_2O$/MWCNT-10% | 2.4 ± 0.3 log | 1.8 ± 0.1 log | 0.5 ± 0.3 log |
| $Cu_2O$/MWCNT-25% | 4.0 ± 0.0 log | 3.2 ± 0.2 log | 1.4 ± 0.2 log |

caused by the different nature of initial precursor applied (notably, inorganic iron chloride was used over against organic Cu and Ti precursors) and can be an appreciable drawback in virus removal efficiency. However, the particle size of pure α-$Fe_2O_3$ is also the highest compared to either pure titania or $Cu_2O$, thus the specific surface areas of their composites are very low even below $10 \, m^2 \, g^{-1}$. The specific surface areas of $Cu_2O$/MWCNT nanocomposites are just somewhat lower than that of pristine MWCNT (table 1) which can also have a positive effect on remarkable virus removal capacity.

## 3.2. Virus removal with MWCNT-based nanocomposite hybrid membranes

### 3.2.1. Batch experiments

The results of batch experiments at different pH values for the nanocomposite and raw materials are presented in table 2. As we have not found significant differences between the samples collected between $t = 1$ min and $t = 5$ min, all of the presented virus retention values show the last sampling point ($t = 5$ min). From these data, it is obvious that the virus removal efficiency fluctuates significantly with varying pH. It was found that $Cu_2O$-coated MWCNT nanocomposites provided the most promising results in the examined pH range. Comparing the nanocomposites containing various

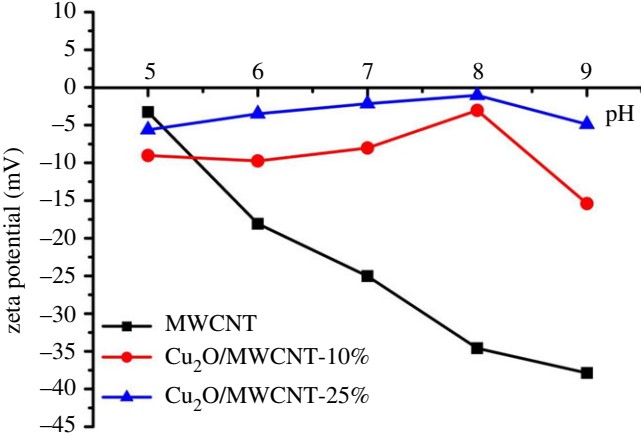

**Figure 6.** Zeta potential as a function of pH for pristine MWCNT (black curve) and Cu$_2$O/MWCNT-based nanocomposites (blue and red curves).

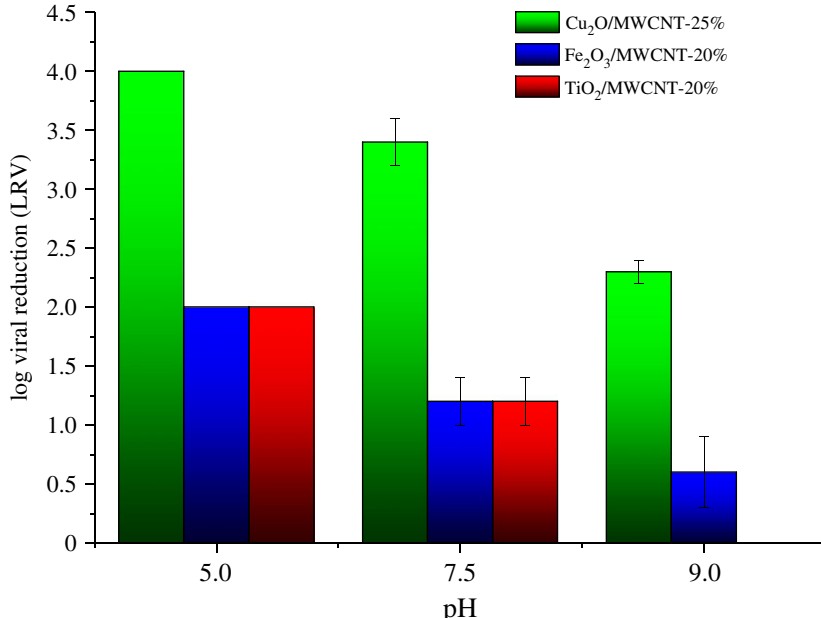

**Figure 7.** MS2 retention of the MWCNT-based nanocomposite membranes in flow experiments applying 150 dm$^3$ m$^{-2}$ h$^{-1}$ water flux.

inorganic materials, we have found that only Cu$_2$O/MWCNT samples showed virus retention properties at pH = 9 (figure 5). Based on these results, adsorption tests with higher MS2 concentration (2 ml MS2 of the $5 \times 10^6$ PFU ml$^{-1}$—LRV ≤ 4-Log) were also performed on Cu$_2$O/MWCNT nanocomposites (table 2 and figure 5). While the Cu$_2$O/MWCNT nanocomposites showed LRVs of up to 1.4 at pH 9, 3.2 at pH 7.5 and at least 4.0 at pH 5.0, respectively (table 2 and figure 5), the LRVs of TiO$_2$- and the Fe$_2$O$_3$-coated MWCNT materials revealed that these nanocomposites did not influence the virus retention appreciably.

As it was previously highlighted, during virus filtration not only the specific surface area but also the zeta potential ($\zeta$) values of adsorbents are very significant parameters for virus rejection. Virus retention can be explained by two main issues in our system: the inactivation of virions and their surface adsorption. To better understand the ongoing mechanism, $\zeta$ potential measurements were carried out on the nanocomposites, while we used the literature data for MS2 [54]. Prior to the experiments, the samples were, respectively, dispersed in VDB solution to reach a final concentration of 0.2 wt%, and the pH was adjusted by adding either HCl or NaOH solution. Figure 6 shows the $\zeta$ potential of MWCNTs, Cu$_2$O/MWCNT-10% and Cu$_2$O/MWCNT-25%, respectively, in the pH range of 5.0–9.0.

The two different profiles can be easily distinguished: while the pure MWCNT sample shows a continuously decreasing character, the $\zeta$ potential of the composites increases with increasing pH up to pH = 8 and then changes to a decaying tendency. This indicates that the electrostatic properties of the

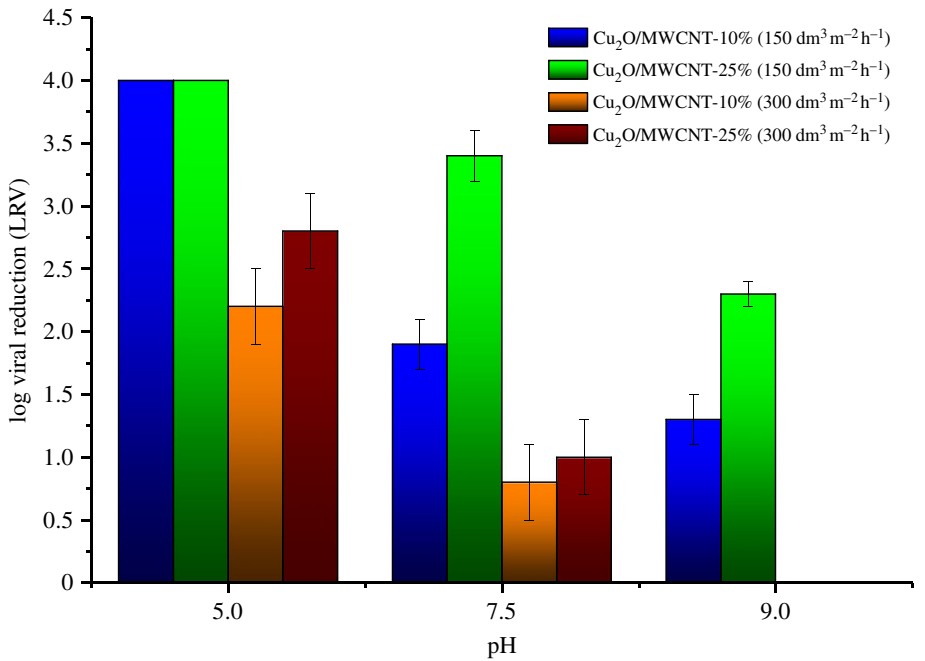

**Figure 8.** MS2 retention of the Cu$_2$O/MWCNT nanocomposite membranes in flow experiments applying 150 dm$^3$ m$^{-2}$ h$^{-1}$ and 300 dm$^3$ m$^{-2}$ h$^{-1}$ water fluxes.

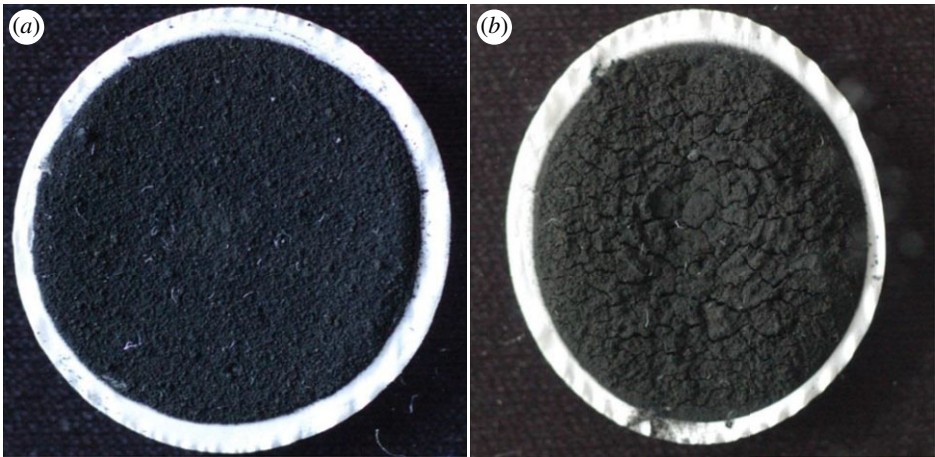

**Figure 9.** Representative photographs of Cu$_2$O/MWCNT-25% membranes after flow experiments at pH 7.5 applying 150 dm$^3$ m$^{-2}$ h$^{-1}$ (*a*) and 300 dm$^3$ m$^{-2}$ h$^{-1}$ (*b*) flow rates.

MWCNTs are favourably influenced for virus retention by being decorated with Cu$_2$O. At pH = 5, the $\zeta$ potential of the Cu$_2$O-covered MWCNT samples is more negative than that of pure MWCNTs, yet the LRVs are higher, which is in accordance with [32] stating that the Cu$_2$O patches on the surface participate in other virus inactivation mechanisms as well. Another interesting phenomenon is the less negative zeta potential of Cu$_2$O/MWCNT-25%, compared with Cu$_2$O/MWCNT-10%. As MWCNT possesses a strongly negative $\zeta$ potential, one could expect the values to show the exact opposite profile. However, considering the specific surface area data in table 1, the sample with higher MWCNT content possesses a 13% higher surface area. The $\zeta$ potential arises from the surface charge density, which is inversely proportional to the surface area, and supposing that the Cu$_2$O coverage of the MWCNTs does not change in the two samples, the higher surface area of the sample results in an overall less negative average $\zeta$ potential.

### 3.2.2. Flow experiments

Based on the promising results of batch experiments, flow experiments were performed with samples of increased MWCNT content (20 and 25 wt%). Furthermore, as Cu$_2$O/MWCNT nanocomposites provided

**Table 3.** MS2 bacteriophage removal efficiency of MWCNT-based nanocomposite hybrid membranes at varying pH and water flux values in flow experiments.

| sample | LRV—pH 5.0 | | LRV—pH 7.5 | | LRV—pH 9.0 | |
| | $F$ (150 dm$^3$ m$^{-2}$ h$^{-1}$) | $F$ (300 dm$^3$ m$^{-2}$ h$^{-1}$) | $F$ (150 dm$^3$ m$^{-2}$ h$^{-1}$) | $F$ (300 dm$^3$ m$^{-2}$ h$^{-1}$) | $F$ (150 dm$^3$ m$^{-2}$ h$^{-1}$) | $F$ (300 dm$^3$ m$^{-2}$ h$^{-1}$) |
|---|---|---|---|---|---|---|
| TiO$_2$/MWCNT 20% | 2.0 ± 0.0 log | 2.0 ± 0.0 log | 1.2 ± 0.2 log | 0.2 ± 0.1 log | 0.0 ± 0.0 log | 0.0 ± 0.0 log |
| Fe$_2$O$_3$/MWCNT 20% | 2.0 ± 0.0 log | 2.0 ± 0.0 log | 1.2 ± 0.4 log | 0.3 ± 0.2 log | 0.6 ± 0.3 log | 0.0 ± 0.0 log |
| Cu$_2$O/MWCNT 10% | 4.0 ± 0.0 log | 2.2 ± 0.3 log | 1.9 ± 0.2 log | 0.8 ± 0.3 log | 1.3 ± 0.2 log | 0.0 ± 0.0 log |
| Cu$_2$O/MWCNT 25% | 4.0 ± 0.0 log | 2.8 ± 0.3 log | 3.4 ± 0.2 log | 1.0 ± 0.3 log | 2.3 ± 0.1 log | 0.0 ± 0.0 log |

the highest virus retention values, a $Cu_2O$/MWCNT composite sample with decreased MWCNT content (10%) was also investigated in flow experiments. To test the effects of the flow rate, all experiments were carried out at two different water flux values ($150\,dm^3\,m^{-2}\,h^{-1}$ (flow rate $5\,ml\,min^{-1}$) and $300\,dm^3\,m^{-2}\,h^{-1}$ (flow rate $10\,ml\,min^{-1}$)) using different nanocomposite hybrid membranes (figures 7 and 8). The results reassured that there is a high degree of similarity with the observations under batch conditions. Considering the whole examined pH range, it was found that the $Cu_2O$/MWCNT-25% nanocomposite ensured the highest adsorption values at both water flux values. Figure 7 shows LRV of 4.0 at pH 5.0, 3.4 at pH 7.5 and 1.7 at pH 9, respectively, with a water flux of $150\,dm^3\,m^{-2}\,h^{-1}$ for the $Cu_2O$/MWCNT-25% nanocomposite membrane. As presented in figure 7 and table 3, nanocomposite membranes containing 25 wt% MWCNT showed better virus retention capability than those with 10 wt% MWCNT.

In accordance with the batch experiments and $\zeta$ potential measurements, the increased MWCNT content had a positive impact on the virus removal efficiency, most probably due to the higher average surface area of the sample. Similarly to batch experiments, $TiO_2$- and $Fe_2O_3$-coated MWCNTs did not show significant performance in the virus retention in flow experiments either. However, it is worth considering that the LRVs were definitely higher in flow experiments than in batch experiments using the same nanocomposite. As discussed in our previous work [40], nanocomposite membranes with elongated particles, such as MWCNT, have a complex three-dimensional structure, consequently, the contact time of MS2 bacteriophages and the $Cu_2O$ particles is longer, which can yield higher LRVs.

When the higher water flux was applied ($300\,dm^3\,m^{-2}\,h^{-1}$), membrane damage was observed in many cases. Figure 9 shows representative pictures of membranes after filtration used in flow experiments. As can be clearly seen, cleavages occur in the membrane surface at higher flux ($300\,dm^3\,m^{-2}\,h^{-1}$). It was supposed that the increased pressure caused severe structure damage in the MWCNT-containing hybrid membranes during filtration, which resulted in decreased virus removal efficiency via unfavourable shortcuts. In other words, it can be explained by the fact that when a bacteriophage passes through a crack in a membrane, the electrostatic attractions are not sufficient to attract it to the adsorbent due to the large distance.

# 4. Conclusion

In this study, a successful attempt with MWCNT-based nanocomposite hybrid membrane was presented, which could provide a new technology pathway for water purification. The overall excellent performance of the $Cu_2O$/MWCNT nanocomposite membranes for virus removal suggests that further development of the produced filters is of great promise for the powerful treatment of virus-contaminated water. The LRVs were investigated both under batch and flow conditions in the extended pH range of natural waters. Experiments revealed that the $Cu_2O$/MWCNT membranes provide noteworthy virus retention capability and our results confirmed a virus retention of up to 4-Log (99.99%) and possibly even higher, which only slightly decreased approaching the neutral pH value. Hence, efficiency as 'at least' LRV = 4.00 was indicated because there were no more bacteriophages to be removed at the end of the experiment. In other words, it can mean that the effectiveness of the membranes can be even higher. Future experiments with higher initial virion concentrations are planned to judge the real LRV for the selected nanocomposite. Also, strong conclusions can be drawn about the indisputable effect of the presence of MWCNT on the virus retention, because higher MWCNT content resulted in an increase in the LRVs. The positive role of MWCNT in virus removal mechanism can be explained either by the higher surface area or their special surface properties (compared to inorganic particles), thus presenting higher adsorption capacity, which can be advantageous for virion uptake.

Comparing inorganic components of nanocomposite materials, $Cu_2O$-containing samples showed the highest efficiency in virus filtration at all pH values. Moreover, via probable synergetic effect, MWCNT-based nanocomposite hybrid membranes proved to be the hopeful solution for environmental cleaning, because MWCNT could potently increase adsorption efficacy of organic pollutants from water and also serve as high-surface-area support for $Cu_2O$-based virus adsorbent. In the future, we would like to perform stability and adsorption tests on the membranes after their optimization to larger water quantities, in order to provide a modern alternative for everyday virus filtration in general household applications.

Data accessibility. Our data are deposited at Dryad: http://dx.doi.org/10.5061/dryad.r2f59f0 [55].
Authors' contributions. Z.N. performed membrane preparation and filtration experiments. G.P.S. optimized the inorganic functionalization and helped in the filtration experiments. M.S. helped in optimizing the virus filtration experiments.

K.S. addressed the analytical chemical problems in the work. J.T. helped in handling microbiological samples. W.P., K.H. and T.G. contributed to conceiving of the study, designing the study and in its coordination. All authors contributed to the preparation of the manuscript, and all authors accepted the final version of it.

Competing interests. The authors declare no competing interests.

Funding. Financial support came from the Swiss Contribution—Scientific Exchange Programme (Sciex-project no. 14.119).

Acknowledgements. The work was supported by the Swiss Contribution—Scientific Exchange Programme (Sciex-project no. 14.119). Z.N. acknowledges the support of the European Union and the Hungarian Government in the framework of the GINOP 2.3.4-15-2016-00004 'Advanced materials and intelligent technologies to promote the cooperation between the higher education and industry'. We are thankful for the help of Dr Balázs Réti in the Raman spectroscopy measurements.

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
