## [Reviewer comments · Royal Society Open Science]

Review History

RSOS-181294.R0 (Original submission)

Review form: Reviewer 1

Is the manuscript scientifically sound in its present form?

Yes

Are the interpretations and conclusions justified by the results?

Yes

Is the language acceptable?

Yes

Is it clear how to access all supporting data?

Yes

Do you have any ethical concerns with this paper?

No

Have you any concerns about statistical analyses in this paper?

No

Recommendation?

Accept as is

Comments to the Author(s)

The comment is given in the report (see Appendix A).

Review form: Reviewer 2

Is the manuscript scientifically sound in its present form?

Yes

Are the interpretations and conclusions justified by the results?

Yes

Is the language acceptable?

Yes

Is it clear how to access all supporting data?

No

Do you have any ethical concerns with this paper?

No

Have you any concerns about statistical analyses in this paper?

No

Recommendation?

Major revision is needed (please make suggestions in comments)

Comments to the Author(s)

The article presents an interesting methodology for the synthesis of membranes, but some changes must be made as indicated below:
"Sigma-Aldrich" does not exist.

In the formula of Flux (F) superindices must be used.

Reference 2 is incorrect.

Reference 5 is incorrect.

Reference 7 is incorrect.

Reference 11 is incorrect.

Reference 16 is incorrect.

Reference 17 is incorrect.

Reference 18 is incorrect.

Reference 19 is incorrect.

Reference 25 is incomplete.

Reference 26 is incorrect.

Reference 33 is incorrect.
Reference 34 is incorrect.
Reference 35 is incorrect.
Reference 36 is identical to the reference 1.
Reference 40 is identical to the reference 7.
Reference 42 is incorrect.
Reference 43 is incomplete.
Reference 44 is incorrect.
Reference 53 is incorrect, use subindices.

Decision letter (RSOS-181294.R0)

26-Oct-2018

Dear Professor Hernadi:

Title: Enhanced virus filtration in hybrid membranes with MWCNT nanocomposite
Manuscript ID: RSOS-181294

The editor assigned to your manuscript has now received comments from reviewers. We would like you to revise your paper in accordance with the referee and Subject Editor suggestions which can be found below (not including confidential reports to the Editor). Please note this decision does not guarantee eventual acceptance.

Please submit your revised paper before 18-Nov-2018. Please note that the revision deadline will expire at 00.00am on this date. If we do not hear from you within this time then it will be assumed that the paper has been withdrawn. In exceptional circumstances, extensions may be possible if agreed with the Editorial Office in advance. We do not allow multiple rounds of revision so we urge you to make every effort to fully address all of the comments at this stage. If deemed necessary by the Editors, your manuscript will be sent back to one or more of the original reviewers for assessment. If the original reviewers are not available we may invite new reviewers.

On behalf of the Subject Editor Professor Anthony Stace and the Associate Editor Professor Claire Carmalt.

RSC Associate Editor:
Comments to the Author:
(There are no comments.)

RSC Subject Editor:
Comments to the Author:
(There are no comments.)

Reviewers' Comments to Author:
Reviewer: 1

Comments to the Author(s)
The comment is given in the report.

Reviewer: 2

Comments to the Author(s)
The article presents an interesting methodology for the synthesis of membranes, but some changes must be made as indicated below:
"Sigma-Aldrich" does not exist.
In the formula of Flux (F) superindices must be used.
Reference 2 is incorrect.
Reference 5 is incorrect.
Reference 7 is incorrect.
Reference 11 is incorrect.
Reference 16 is incorrect.
Reference 17 is incorrect.
Reference 18 is incorrect.
Reference 19 is incorrect.
Reference 25 is incomplete.
Reference 26 is incorrect.
Reference 33 is incorrect.

Reference 34 is incorrect.
Reference 35 is incorrect.
Reference 36 is identical to the reference 1.
Reference 40 is identical to the reference 7.
Reference 42 is incorrect.
Reference 43 is incomplete.
Reference 44 is incorrect.
Reference 53 is incorrect, use subindices.

Author's Response to Decision Letter for (RSOS-181294.R0)

See Appendix B.

Decision letter (RSOS-181294.R1)

13-Nov-2018

Dear Professor Hernadi:

Title: Enhanced virus filtration in hybrid membranes with MWCNT nanocomposite
Manuscript ID: RSOS-181294.R1

It is a pleasure to accept your manuscript in its current form for publication in Royal Society Open Science. The chemistry content of Royal Society Open Science is published in collaboration with the Royal Society of Chemistry.

On behalf of the Subject Editor Professor Anthony Stace and the Associate Editor Professor Claire Carmalt.

RSC Associate Editor
Comments to the Author:
(There are no comments.)

Reviewer(s)' Comments to Author:

Appendix A

Referee report on the ms “Enhanced virus filtration in hybrid membranes with MWCNT nanocomposite” by Nemeth et al.

The work deals with fabrication and testing of carbon nanotubes (CNT) based composite filtration membrane for removing viruses from water. The filter consists of polytetrafluoroethylene (PTFE) and Multi-walled CNT decorated with various nanoparticles. The virus retention (MS2 bacteriophages) scaled with the % of CNTs and depended on the pH, as well. The best results were obtained with Cu₂O-containing samples.

The starting materials and the filter were thoroughly characterized (x-rays, TEM, Raman, zeta potential), the parameters were varied in a broad range, there are no loose ends. The literature is well cited and the manuscript is very well written, the figures are nice. I am impressed by this work, it is one of the best I have read in this topic. I can fully support the publication of this work.

A follow-up work could be to test the filter performance under illumination. My guess is that the efficiency would increase, the number of “vital” MS2 bacteriophages would decrease, due to the hyperthermia pasteurization coming from the heating up of the CNTs with light, and the photocatalytic effect of nanoparticles. It is very likely, that filters decorated with TiO₂

Appendix B

Response to the comments of Reviewer 1

We thank Reviewer 1 for their careful reading of our manuscript. We are grateful for the positive feedback.

Reviewer 1: "A follow-up work could be to test the filter performance under illumination. My guess is that the efficiency would increase, the number of "vital" MS2 bacteriophages would decrease, due to the hyperthermia pasteurization coming from the heating up of the CNTs with light, and the photocatalytic effect of nanoparticles."

Author's response: We fully agree with Reviewer 1 on this matter. This proposition could potentially enhance the performance of our filter by several magnitudes, since more characteristics of the nanotubes would be exploited. Similar studies have already been conducted in our laboratory, and further experiments will focus on the illumination of the samples.

Response to the comments of Reviewer 2

We are grateful for the comments of Reviewer 1 for his/her careful reading of our manuscript, and their detailed comments that we have addressed. The modifications based on his/her comments greatly improved the manuscript.

Reviewer 2: "Sigma-Aldrich does not exist".

Author's reply: We agree that changes have been in place in the company structures, which today would suggest us to use Merck as distributor. However, the chemicals used in these experiments were ordered from the original Sigma-Aldrich company and thus the source of chemicals is correct as written.

Reviewer 2: "In the formula of Flux (F) superindices must be used."

Author's reply: The formula has been corrected accordingly.

Reviewer 2: comments on reference formats

Author's reply: We thank the Reviewer for his/her careful consideration of the Reference list, and the detailed description of mistakes in the references. We used the tools proposed on the website for reference editing, and all the points have been addressed in the revised manuscript.